# The Presence of Caffeic Acid in Cerebrospinal Fluid: Evidence That Dietary Polyphenols Can Cross the Blood-Brain Barrier in Humans

**DOI:** 10.3390/nu12051531

**Published:** 2020-05-25

**Authors:** Izabela Grabska-Kobylecka, Justyna Kaczmarek-Bak, Malgorzata Figlus, Anna Prymont-Przyminska, Anna Zwolinska, Agata Sarniak, Anna Wlodarczyk, Andrzej Glabinski, Dariusz Nowak

**Affiliations:** 1Department of Clinical Physiology, Medical University of Lodz, 92-215 Lodz, Poland; izabela.grabska-kobylecka@umed.lodz.pl; 2Department of Neurology and Stroke, Medical University of Lodz, 90-549 Lodz, Poland; justynakaczmarek22@gmail.com (J.K.-B.); mfiglus@gmail.com (M.F.); andrzej.glabinski@umed.lodz.pl (A.G.); 3Department of General Physiology, Medical University of Lodz, 92-215 Lodz, Poland; anna.przyminska@umed.lodz.pl (A.P.-P.); agata.sarniak@umed.lodz.pl (A.S.); 4Cell-to-Cell Communication Department, Medical University of Lodz, 92-215 Lodz, Poland; anna.zwolinska@umed.lodz.pl; 5Department of Sleep Medicine and Metabolic Disorders, Medical University of Lodz, 92-215 Lodz, Poland; anna.wlodarczyk@umed.lodz.pl

**Keywords:** caffeic acid, plant phenolics, cerebrospinal fluid, blood-brain barrier, dietary polyphenols

## Abstract

Epidemiological data indicate that a diet rich in plant polyphenols has a positive effect on brain functions, improving memory and cognition in humans. Direct activity of ingested phenolics on brain neurons may be one of plausible mechanisms explaining these data. This also suggests that some phenolics can cross the blood-brain barrier and be present in the brain or cerebrospinal fluid. We measured 12 phenolics (a combination of the solid-phase extraction technique with high-performance liquid chromatography) in cerebrospinal fluid and matched plasma samples from 28 patients undergoing diagnostic lumbar puncture due to neurological disorders. Homovanillic acid, 3-hydroxyphenyl acetic acid and caffeic acid were detectable in cerebrospinal fluid reaching concentrations (median; interquartile range) 0.18; 0.14 µmol/L, 4.35; 7.36 µmol/L and 0.02; 0.01 µmol/L, respectively. Plasma concentrations of caffeic acid (0.03; 0.01 µmol/L) did not correlate with those in cerebrospinal fluid (ρ = −0.109, *p* = 0.58). Because food (fruits and vegetables) is the only source of caffeic acid in human body fluids, our results indicate that the same dietary phenolics can cross blood-brain barrier in humans, and that transportation of caffeic acid through this barrier is not the result of simple or facilitated diffusion.

## 1. Introduction

Polyphenols existing in fruits and vegetables represent a group of chemical substances that are characterized by the presence of one or more phenol rings per molecule. The most common and important low molecular weight phenolic compounds are simple phenolic derivatives (phenolic acids e.g., caffeic acid, coumaric acid, gallic acid, hydroxycinnamic acids) and flavonoids (e.g., quercetin, catechin, cyanidin, pelargonin). Phenolic acids account for about one-third of the total dietary intake of polyphenols: flavonoids account for the remaining two-thirds [1]. They have various biological properties, including anti-inflammatory, anti-cancer and anti-oxidant activities [1,2,3,4]. Epidemiological studies revealed inverse relationship between the risk of cardiovascular diseases, malignant neoplasm, diabetes, osteoporosis and the consumption of a polyphenolic rich diet [3,4]. Moreover, numerous observational and interventional studies showed positive effect of increased polyphenols dietary intake on cognitive performance in older subjects [2]. For instance, in the group of 2031 subjects aged 70–74 years, consumers of chocolate, wine or tea (products rich in plant polyphenols) had higher mean cognitive test scores than non-consumers [5]. Low flavonoid consumers without dementia revealed higher decline in cognitive performance over 10 years’ follow up [6]. Supplementation of daily diet with complex antioxidant blend composed of plant extracts, including grape seed, ginkgo biloba and gotu cola, for 4 months in seniors without dementia resulted in the significant improvement of memory [7]. An improvement of spatial working memory was also observed in older adults after treatment with flavonoid antioxidant Pycnogenol in a daily dose 150 mg for 3 months [8]. One of the mechanisms which can explain benefits of high dietary intake of polyphenols for brain function could be a direct effect of this plant compound on neurons, involving the suppression of neuro-inflammation, decrease in stress signal transduction pathways, or expression of genes encoding neuro-protective and neurotrophic proteins [9]. To evoke such action, circulating polyphenols and their metabolites have to cross the blood-brain barrier (BBB) or blood-cerebrospinal fluid (CSF) barrier [10,11], be present in the close neighborhood of neurons and interact with membrane or intracellular receptors. These suggest that polyphenols could be detected in the brain tissue and CSF fluid samples. Dietary supplementation with blueberry or blackberry extracts resulted in the appearance of anthocyanins in various regions of the brain in rats [12,13]. Additionally, other polyphenols, such as naringenin, epicatechin and epigallocatechin, were found in the brain of laboratory animals after intravenous or oral administration [14,15,16]. Moreover, green tea polyphenols were able to reduce the elevated BBB permeability, probably by inhibition of caveolin-1 expression in the rat model of experimental cerebral ischemia caused by middle cerebral artery occlusion [17]. Ferulic acid was detected in the CSF of rats and mice [18,19]. Detectable concentrations of vanillic acid, homovanillic acid (HVA), vanillylmandelic acid and 3,4-dihydroxyphenylacetic acid were reported in human CSF samples [20,21,22,23,24] as markers of brain catecholamines (adrenaline, noradrenaline, dopamine) metabolism, but not as indicators of neurons exposition to dietary phenolics. On the other hand, catechin and epicatechin were reported to cross through the model of BBB composed of immortalized human cerebral micro-vessel cell monolayer in vitro. In addition, BBB cells were able to conjugate these polyphenols with glucuronic acid [25]. Recently, the transportation of quercetin-glucosides, kaempferol, myricetin and myricetin glucosides via a similar in vitro BBB model was also described [26]. These results suggest that various phenolics, and not only those related to catecholamine metabolism, could be present in human CSF. Therefore, in this study, we wanted to determine the concentrations of 12 phenolics with a combination of solid-phase extraction technique with high-performance liquid chromatography, with electrochemical or ultraviolet-visible detection in samples of CSF and matched plasma obtained from 28 patients undergoing diagnostic procedures due to neurological disorders. Furthermore, the correlations between CSF phenolics and selected clinical variables were analyzed.

## 2. Materials and Methods

### 2.1. Study Protocol and Patients

Thirty patients on a “western” diet, who have been diagnosed in the Department of Neurology and Stroke, Medical University of Lodz, were recruited in the study (Table 1). The inclusion criteria involved were: age between 20 and 80 years and a written informed consent before initiating the study procedures. The exclusion criteria included: stroke, head trauma, acute liver failure, hypertensive encephalopathy, HIV infection, alcohol and illicit drug abuse and any history of intestinal malabsorption disorders. Blood and CSF samples for polyphenols measurement were collected under fasting conditions (about 15 h after last meal, i.e., supper) between 8:30 and 9:30 a.m. The study was conducted in accordance with the Declaration of Helsinki. The Medical University of Lodz Ethics Committee approved the study protocol (approval RNN/43/14/KE).

### 2.2. Cerebrospinal Fluid Collection

One and a half of the milliliter cerebrospinal fluid samples (CSF) were collected by lumbar puncture in polypropylene tubes. Half a milliliter thereof was placed into a separate Eppendorf tube, centrifuged (10 min, 1500× *g*, 4 °C) and the supernatant was stored at −80 °C until the phenolics measurement, but not for longer than 3 months. The rest was used for the determination of routine parameters. CSF samples with red blood cells count >500/µL were recognized as bloody [27] and excluded, in order to avoid false positive results of phenolics determination.

## 3. Matched Plasma Samples

Four mL fasting venous blood was collected into vacutainer tubes with EDTA (Becton Dickinson, Franklin Lakes, NJ, USA), within 30 min before lumbar puncture. Blood was centrifuged (10 min, 1500× *g*, 4 °C) and obtained plasma was stored under the same conditions as CSF samples.

### 3.1. Determination of Selected Phenolics in CSF and Plasma Samples

We determined 12 phenolics in CSF and plasma specimens with a combination of solid-phase extraction technique with high-performance liquid chromatography, with electrochemical (HPLC-ECD) or ultraviolet-visible (HPLC-UV-Vis, diode array detector wavelengths 210, 280, and 325 nm) detection, as previously described [28,29]. Vanillic acid, dihydrocaffeic acid, caffeic acid (CA), and HVA were determined by HPLC-ECD, while 3-hydroxyphenyl acetic acid (3HPAA), hippuric acid, 3-hydroxyhippuric, 4-hydroxyhippuric, 3,4-dihydroxybenzoic acid, chlorogenic acid, ellagic acid, and urolithin A were determined by HPLC-UV-Vis. All procedures related to polyphenols determination were the same for CSF and plasma samples. The instrumentation of HPLC-ECD and HPLC-UV-Vis and chromatographic conditions and parameters were described in detail elsewhere [28,29]. All chemicals, buffers, and water used throughout the study were of high-performance liquid chromatography (HPLC) grade.

### 3.2. Solid Phase Extraction Technique

Plasma or CSF samples (0.5 mL) were mixed with 50 μL of 0.78 mol/L acetate buffer (pH 5.2), 25 μL of 20% ascorbic acid solution, 10 μL of 2 mg/L fisetyn solution, 10 μL of *β*-glucuronidase from bovine liver (type B-3) solution in 0.1 mol/L acetate buffer (4000 U/mL, pH 5.2), and 10 μL of sulphatase from Helix pomatia (type H-1) solution in 0.1 mol/L acetate buffer (1028 U/mL, pH 5.2; Sigma-Aldrich Chemical, St. Louis, MO, USA), and incubated for 60 min at 37 °C (to hydrolyze phenolic glucuronides and to remove sulfate group from phenolics). Then, 0.5 mL of 1 mol/L phosphoric acid solution was added and the sample was incubated again for 10 min at 37 °C and then mixed with 0.5 mL water and poured into a Speedisk Column H2OPhobic DVB (3 mL solid phase extraction column, 50 mg per column, 15 μm particle diameter, J.T. Baker, Phillipsburg, NJ, USA), preconditioned with 1 mL of methanol and 0.5 mL of water. The column was washed three times with 0.5 mL of water, dried under a vacuum, and then eluted with 0.4 mL of methanol. The methanol eluate containing polyphenols was used for further analyses [28,29].

#### 3.2.1. HPLC-UV-Vis Separation and Detection

Five µl aliquots of methanol eluate were analyzed with gradient elution HPLC-UV-Vis and flow rate of 0.25 mL/min. The mobile phase consisted of solvent A (0.05% aqueous solution of H_3_PO_4_) and solvent B (0.05% solution of H_3_PO_4_ in acetonitrile). The system was run with a gradient program: 4%–15% B (0–8 min), 15%–40% B (15 min), 40%–50% B (6 min), 50% B (1 min), 50%–4% B (1 min), and 4% B (10 min). Separation was achieved with Phenomenex HPLC column (Synergi 4μm Fusion-RP 80 A), and detection was executed with a diode array detector (wavelengths 210, 280, and 325 nm, UVD340 U Dionex, Sunnyvale, CA, USA) [28,29].

#### 3.2.2. HPLC-ECD Separation and Detection

Fifty μL samples of the eluate were mixed with 450 μL of mobile phase and 20 μL samples of these solutions were subjected to isocratic analysis with HPLC-ECD. The flow rate of the mobile phase was kept constant at 0.20 mL/min and the total run time was 90 min. The mobile phase was prepared by addition of 0.75 g KH_2_PO_4_, 40 mg EDTA, 3 mL 1 mol/L H_3_PO_4_, and 45 mL of methanol to 440 mL of water, and then the pH was adjusted to 2.80 with concentrated CH_3_COOH, and the volume was made up to 500 mL with water. Separation was achieved with Hypersil BDS C18 column (150 × 2.1 mm ID, 3-μm particles) and detection was executed with an electrochemical detector, with a flow-through detection cell equipped with a glass carbon electrode and an Ag/AgCl reference one set at +0.82 V; working temperature 32 °C (Decade, Antec Leyden, The Netherlands) [28,29].

#### 3.2.3. HPLC Data Collection and Elaboration

Chromeleon software (Dionex, Sunnyvale, CA, USA) was used for chromatographic data collection and the calculation of phenolics concentrations. Identification of particular compounds was conducted on the basis of their spectral characteristics and retention times in comparison to corresponding standard substances. All standards came from Sigma-Aldrich Chemie GmbH (Steinheim, Germany) or from Fluka, Sigma-Aldrich (Buchs, Steinheim, Germany). Individual results were obtained as means from triplicate measurements and expressed in micromoles per liter. The detection and determination limits and phenolics recovery from solid phase extraction columns were described elsewhere [29].

### 3.3. Other Determinations

Routine analysis of CSF samples and determination of plasma concentrations of C-reactive protein (CRP) were performed in the Diagnostic Laboratory of University Clinical Hospital Military Memorial Medical Academy of Medical University of Lodz.

## 4. Statistical Analyses

Results are expressed as a mean (SD) and median (Me) and interquartile range (IQR). Differences between plasma and CSF concentrations of selected phenolics were evaluated, alongside differences between patient subgroups, such as sclerosis multiplex (SM) vs. other neurological disorders (OND) subgroup. They were estimated with the U Mann–Whitney, due to the non-parametric distribution of data evaluated with Shapiro–Wilk’s W test (STATISTICA 13.1 StatSoft software). Spearman’s rank correlation coefficient (ρ) was used for analysis associations between CSF phenolics levels and other parameters. A *p* value < 0.05 was considered significant.

## 5. Results

Figure 1 and Figure 2 illustrate representative chromatograms of the CSF sample obtained from a single patient. From 12 measured phenolics, only three (CA, HVA and 3HPAA) were detected in both CSF and corresponding plasma samples. However, the number of positive plasma samples for 3HPAA was approximately 5-times lower than the number of positive readings of this compound in CSF samples (5 vs. 27, Table 2). Dihydrocaffeic acid, vanillic acid and hippuric acid were detected in about half of studied plasma samples, while CSF was always negative for these compounds. The remaining phenolics (3-hydroxyhippuric, 4-hydroxyhippuric, chlorogenic acid, ellagic acid, and urolithin A) were detected neither in CSF, nor in plasma samples (Table 2).

### 5.1. CSF and Plasma Concentrations of Caffeic Acid, Homovanillic Acid and 3-hydroxyphenyl Acetic Acid

Figure 3 shows the mutual comparisons of concentrations of CA, HVA, and 3HPAA in CSF and plasma samples of the whole group of studied patients. 3HPAA had the highest mean CSF concentration (7.96 µmol/L), which was about 40- and 400-times higher (*p* < 0.05) than that of HVA and CA, respectively. Mean CSF concentrations of 3HPAA and HVA were 11- and 2-times higher (*p* < 0.05) respectively than those in plasma, while the levels of CA were similar in these body fluids (Figure 3). Patients with multiple sclerosis (MS) (*n* = 15) did not differ from other neurological disorders (OND) subgroup (*n* = 13), in respect to CSF concentrations of the afore-mentioned 3 phenolics (Table 3). Similarly, plasma levels of HVA and CA were similar in these subgroups. In the MS subgroup, plasma HVA and CA were 0.09 ± 0.02 (0.08; 0.03) µmol/L and 0.03 ± 0.01 (0.03; 0.01) µmol/L, while in the OND subgroup, they reached 0.11 ± 0.01 (0.12; 0.07) µmol/L and 0.03 ± 0.01 (0.03; 0.01) µmol/L, respectively.

### 5.2. Correlations of Polyphenols in CSF with Other Measured Variables

CSF levels of CA and HVA acid did not correlate with their concentrations in plasma (Figure 4 and Figure 5). Similarly, CSF concentrations of CA, HVA and 3HPAA did not correlate with other CSF parameters such as cell count, total protein and glucose (ρ ranged from −0.243 to 0.285, *p* > 0.05).

## 6. Discussion

We found three phenolics: HVA, 3HPAA and CA in almost all studied CSF samples (Appendix A). The presence of CA in human CSF is described for the first time to the best of our knowledge. These afore-mentioned, relatively simple phenolic acids are present in some fruits and vegetables and can also be produced by gut microflora from ingested, more complex polyphenols and then absorbed into the blood stream [30]. Their existence in CSF suggests that they may cross BBB or blood-CSF barrier and be proof that plant phenolics can exert a direct neuro-protective effect explaining epidemiological data on the beneficial influence of a polyphenol rich diet on brain function during ageing [11]. However, this idea has important limitations, because HVA and 3HPAA can be formed endogenously in the brain [31,32,33]. On the other hand, no metabolic pathways leading to endogenous production of CA were identified in the human body so far [34,35]. Thus, CA may confirm the hypothesis that some ingested plant phenolics can reach brain neurons in humans.

### 6.1. Homovanillic Acid in CSF

Certain fractions of catecholamines (adrenaline, noradrenaline, dopamine) released by brain neurons undergoes several reactions catalyzed by brain monoamine oxidase (MAO) and catechol-O-methyltransferase (COMT), resulting in the formation of 3,4-dihydroxyphenylacetic acid, vanillylmandelic acid and HVA [31,32]. Catecholamines produced outside of the central nervous system are metabolized in a similar way [31]. Thus, the afore-mentioned compounds were found in plasma, urine, and CSF samples [22,31,32]. In our study, we measured 12 phenolics, including HVA in CSF and matched plasma samples. Measurement of HVA served as a positive control of our analytical techniques. We found HVA in all samples of CSF and venous plasma. Higher concentration of HVA in CSF than in plasma corresponds to previous observations concerning higher levels of this compound in internal jugular venous plasma than in arterial plasma in humans [36]. Surprisingly, we did not observe significant correlation between CSF and plasma HVA concentrations in studied patients. Although HVA can leak from brain into systemic circulation, the brain derived HVA contributes only to about 10% of total body HVA production [36]. Moreover, dietary flavanols (e.g., quercetin) present in vegetables (e.g., tomato, onion) and beverages (e.g., tea or coffee) can increase the circulating pool of HVA [37,38]. This may explain the negative result of the afore-mentioned correlation analysis. On the other hand, it is interesting whether dietary intervention elevating the HVA circulatory level above that present in CSF can reverse HVA transportation through BBB or the blood-CSF barrier.

### 6.2. 3-hydroxyphenyl Acetic Acid in CSF

Notably, 3HPAA is one of the metabolites of rutin and quercetin formed by colonic microbiota in humans [39,40]. It can be absorbed from the colon into the blood stream, because its urinary excretion rises after the introduction of high polyphenol diet in humans [41]. Increased urinary excretion of this compound was also noted after ingestion of a single dose of soluble cocoa powder or dark chocolate [42,43]. The concentrations of 3HPAA in CSF are similar to those reported by other authors [33,44]. However, the authors recognized 3HPAA as a metabolite of oxidative deamination of p-tyramine [33], and not as a metabolite of plant polyphenols ingestion. p-tyramine belongs to “trace amines” and can act as a neurotransmitter via G-protein coupled receptors in the brain. It can be formed endogenously as a product of enzymatic decarboxylation of tyrosine [45], or ingested with food (e.g., fish, cheese, chocolate, certain vegetables) [46]. In vitro studies with the model of BBB suggest that dietary tyramine can reach the central nervous system [47]. In humans, the diet seems to be the main source of physiologically relevant tyramine concentrations [46]. The mean concentration of 3HPAA was many times higher than that in corresponding plasma. Moreover, the number of 3HPAA positive CSF samples was about 5.4-times greater than the number of positive plasma samples in studied patients. These suggest that CSF 3HPAA could be the result of the penetration of plasma tyramine into the brain, with its subsequent oxidative deamination. Thus the existence of 3HPAA in CSF cannot be recognized as a convincing evidence of any transport of plant phenolics through BBB in humans. Interestingly, increased urinary excretion of 3HPAA was found in children with phenyloketonuria [48] and autistic patients [49], as a consequence of the abnormal metabolism of phenylalanine. The overgrowth of intestinal microbiota such as Clostridium species may also lead to the enhanced formation of 3HPAA [49]. Because some brain dysfunctions are observed in autistic patients and those with untreated phenyloketonuria, one may speculate that circulating 3HPAA may cross BBB and be toxic for neurons. On the other hand, in vitro experiments revealed a protective effect of 3HPAA on the BBB model against oxidative stress [50]. Therefore, as to whether 3HPAA is a biomarker of the afore-mentioned diseases or rather a metabolite involved in the development of brain damage is still an open question.

### 6.3. Caffeic Acid in CSF

We described, for the first time, the existence of CA in human CSF. Food (e.g., olives, fruits, carrots, coffee beans) is the only source of CA in human body fluids. This compound is absorbed in the free form in the colon and undergoes methylation, sulphatation or glucuronidation that makes the molecule of CA more hydrophilic, facilitating its urinary elimination [51]. The concentration of CA in CSF was almost the same as that in plasma. However, there was no correlation between CSF and plasma levels of CA. These suggest that the transportation of CA through BBB could not to be the result of simple or facilitated diffusion. Intestinal cells actively absorb free CA using monocarboxylic acid transporters [51]. Various solute carriers, including monocarboxylic acid transporters, are present on the capillary endothelial cells of BBB [52]. Experiments on animals and on in vitro models of BBB revealed that transportation of various substances including phenolics is affected by their methylation, sulphatation or glucuronidation [53,54]. Methylation or glucuronidation of selected polyphenols increased their traversing through BBB models in vitro [54]. Sulphatation of dopamine caused its permeation through the BBB in rats [53]. Moreover, transient conjugation or de-conjugation of polyphenols can take place in close vicinity of BBB [54]. Since CA is present in the plasma in free and conjugated form, the process of its transportation through BBB may depend on these afore-mentioned factors. These may explain no association between CSF and plasma levels of CA. On the other hand, low concentrations of CA close to the limit of the assay sensitivity can be responsible for the negative results of the correlation analysis. Nevertheless, the presence of CA in CSF shows that at least some dietary plant phenolics after absorption to the blood can cross BBB in humans and directly affect the functions of brain cells. These are in agreement with previous experiments on laboratory animals, showing neuroprotective activity of CA [55,56] and plausible permeation of this phenolic acid, through BBB into brain in mice [57].

## 7. Limitations of the Study

Although exclusion criteria involved diseases (stroke, head trauma, acute liver failure, hypertensive encephalopathy, HIV infection) that destroy the integrity and function of BBB [58,59], other diseases that were an indication for lumbar puncture and CSF collection may also enhance BBB permeability. For instance, MS and diabetes mellitus were reported to increase BBB permeability [58]. Moreover, two patients suffered from meningitis and one from brain tumor, diseases which potentially induce BBB dysfunction. Thus one may suppose that our results concerning CA concentrations in CSF were higher than those in healthy subjects. On the other hand, 4 patients with mononeuropathy or polyneuropathy without coexisting diabetes mellitus and any pathology localized in the brain represented detectable CA in CSF. Systemic inflammation negatively affects BBB function [58]. The majority of studied patients (especially those with MS) had plasma CRP, a marker of intensity of inflammatory response within normal range. In addition, the lack of correlation between CSF and plasma concentrations of CA suggests that simple permeation was not the main mechanism of CA transportation through BBB. These indicate that our results were not significantly biased by the presence of neurological disorders in studied subjects.

We used relatively nonspecific analytical methods (HPLC-UV-Vis and HPLC-ECD) for the measurement of phenolics in human CSF and matched plasma. An application of HPLC with mass spectrometric detection would probably give an opportunity to find more dietary phenolics in CSF. However, despite this, we achieved our goal. We describe for the first time the presence of CA in human CSF. Plasma and CSF samples were collected under fasting conditions after 15 h from the last meal. Maximal increase in CA plasma levels was observed within 0.5–1 h after consumption of 200 mL of red wine containing 1.8 mg of CA [60]. Similarly, an ingestion of 200 mL of instant coffee resulted in the maximal increase in chlorogenic acid plasma levels (an ester of caffeic acid and quinic acid) after 1 h, with a half-life time of about 0.3 h [34]. These suggest that non-fasting plasma and CSF samples could have considerably increased CA concentrations. On the other hand, studies with ileostomy patients revealed that almost all the ingested CA dose was absorbed in the small intestine, while in the case of chlorogenic acid it was only one third of the dose [61]. This indicates that the majority of chlorogenic acid from food could reach the colon and be hydrolyzed by colon microbiota into free CA, with its subsequent absorption into the blood. Thus, the absorption of CA can occur even after several hours from the last meal and the conditions of blood and CSF samples collection can have a rather moderate effect on CA measurement results. However, the considerable differences between CSF concentrations of CA and the remaining two phenolics (HVA and 3HPAA) are in line with these data and support, to the same extent, the conclusion that the presence of caffeic acid in CSF results only from its passage through the blood-brain barrier. Other phenolic acids (e.g., ferulic acid, dihydrocaffeic acid, 3,4-dihydroxybenzoic acid) are also rapidly absorbed in the small intestine and reach maximal plasma concentration, within 1 to 2 h after ingestion. Their elimination lasts about a few hours [60]. Similarly, hippuric acids (hippuric acid, 3-hydroxyhippuric, 4-hydroxyhippuric), which are metabolites of various polyphenols, are rapidly excreted with the urine [60]. Therefore, it is possible that the lack of detectable concentrations of these compounds in plasma and CSF samples could result from the relatively long fasting period preceding their collection.

This is in line with our previous observations [29]. In strawberry consumers (500 g of strawberry pulp daily, between 11:00 a.m. and 2:00 p.m., for 30 days) fasting plasma had no detectable ellegic acid and its main metabolite urolithin A, while HPLC analysis of spot morning urine revealed the elevated concentrations of this last compound [29]. Nevertheless, an analysis of post-meal CSF and plasma samples would clearly solve this issue.

## 8. Concluding Remarks

Screening of human CSF samples for 12 selected phenolics with HPLC-UV-Vis and HPLC-ECD revealed the presence of HVA, 3HPAA and CA. The source of the first two compounds in CSF is dual. They can originate from endogenous metabolic processes in the brain and can be absorbed into the blood as derivatives of ingested plant polyphenols produced by gut microflora. However, the only source of CA in human body fluids is food. This indicates that CA absorbed from ingested fruits and vegetables into the blood stream can cross BBB and reach brain cells. This is direct evidence that some plant phenolics can cross BBB in humans.

## Figures and Tables

**Figure 1 nutrients-12-01531-f001:**
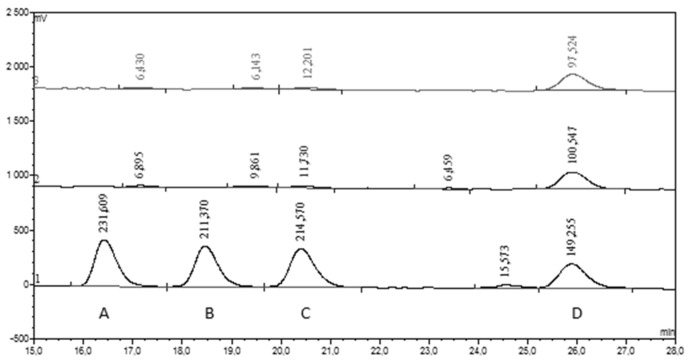
Chromatogram high-performance liquid chromatography, with electrochemical detection (HPLC-ECD). Double chromatographic analysis (HPLC-ECD) of cerebrospinal fluid sample obtained from patient No 9 (Appendix A
Appendix A). The bottom chromatogram illustrates the elution of standards: A—dihydrocaffeic acid, B—vanillic acid, C—caffeic acid, D—homovanillic acid. Numbers over the peaks represent their area.

**Figure 2 nutrients-12-01531-f002:**
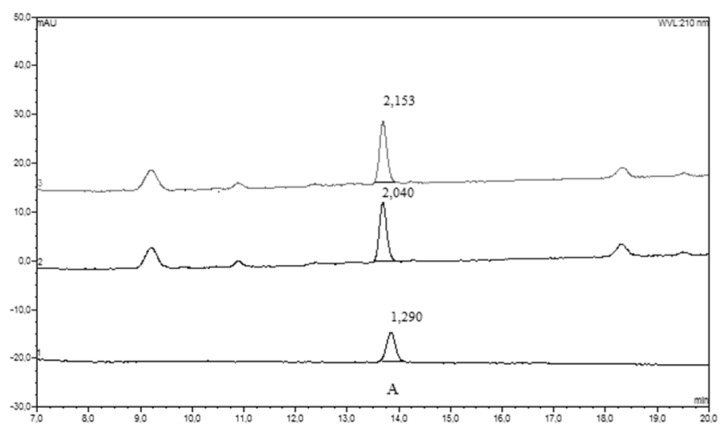
Chromatogram HPLC–UV-Vis. Double chromatographic analysis (HPLC-UV-Vis) of cerebrospinal fluid sample obtained from patient No 9 (Appendix A
Appendix A). The bottom chromatogram illustrates the elution of standards: A—3-hydroxyphenyl acetic acid. Numbers over the peaks represent their area.

**Figure 3 nutrients-12-01531-f003:**
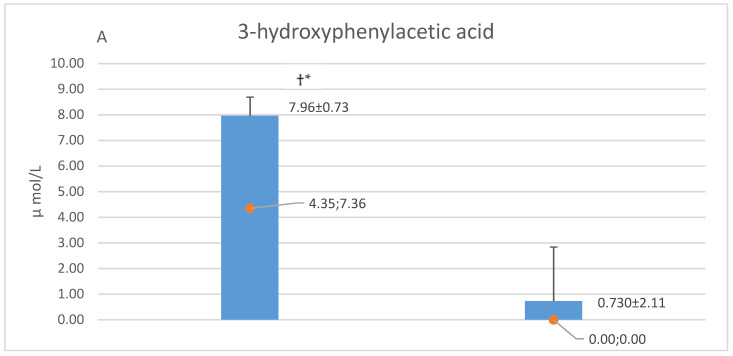
Concentrations of 3-hydroxyphenyl acetic acid (**A**), homovanillic acid (**B**) and caffeic acid (**C**) in cerebrospinal fluid (CSF) and matched plasma samples obtained form 28 studied patients. Results are expressed as mean ± SD and median; IQR (interquartile range). * versus corresponding value in plasma, *p* < 0.05. † versus blood-cerebrospinal fluid (CSF) concentrations of homovanillic acid and caffeic acid, *p* < 0.05. ‡ versus CSF concentration of caffeic acid, *p* < 0.05.

**Figure 4 nutrients-12-01531-f004:**
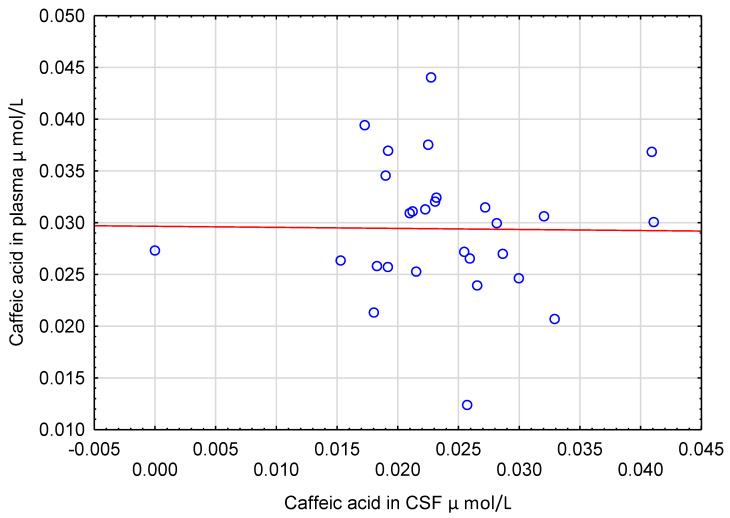
Scatter plot of caffeic acid concentrations in plasma (Y) and cerebrospinal fluid (X). No significant correlation was found between these two variables (ρ = −0.109, *p* = 0.58, *n* = 28).

**Figure 5 nutrients-12-01531-f005:**
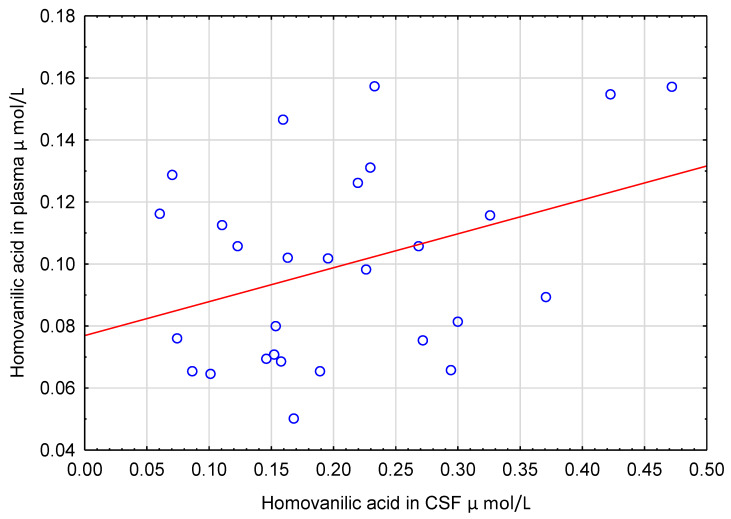
Scatter plot of homovanillic acid concentration in plasma (Y) and cerebrospinal fluid (X). No significant correlation was found between these two variables (ρ = 0.272, *p* = 0.161, *n* = 28). CSF—cerebrospinal fluid.

**Table 1 nutrients-12-01531-t001:** Basic characteristic of studied patients with neurological disorders and samples of cerebrospinal fluid.

Demographic/Clinical Variables	Whole Group ^†^	Multiple Sclerosis	Other Neurological Disorders ^‡^
Number of subjects	28	15	13
Sex F/M	18/10	13/2	5/8
Age [years]	46 ± 16 (44; 29)	40 ± 14 (23; 4)	53 ± 15 (30; 9) *
plasma CRP [mg/L]	4.1 ± 6.1 (1.5; 5.2)	1.7 ± 1.7 (0.2; 0.2)	7.5 ± 8.2 (0.5; 1.0) *
Cerebrospinal fluid samples			
Cell count (cells/µL)	22 ± 41 (6; 9)	10 ± 11 (2; 1)	31 ± 52 (1; 0)
Erythrocytes (cells/µL)	118 ± 127 (3; 74)	123 ± 199 (4; 82)	39 ± 59 (1; 35)
Total protein (g/L)	0.56 ± 0.36 (0.45; 0.19)	0.44 ± 0.12 (0.29; 0.04)	0.65 ± 0.38 (0.30; 0.12)
Glucose (mg/dL)	74 ± 22 (68; 13)	67 ± 6 (69; 7)	81 ± 29 (68; 34)

^†^ initially, thirty patients were recruited, however, two of them dropped out due to the bloody CSF samples. ^‡^ 4 patients had polyneuropathy, 3 mononeuropathy, 2 meningitis, 1 brain tumor, 1 epilepsy, 1 amyotropic lateral sclerosis and 1 suffered from Friedreich’s ataxia. Three patients with polyneuropathy also had diabetes mellitus. * *p* < 0.05 versus sclerosis multiplex subgroup.

**Table 2 nutrients-12-01531-t002:** Number of cerebrospinal fluid and plasma samples with detectable levels of studied polyphenols.

Phenolic Acid	Whole Group *n* = 28			
	Cerebrospinal Fluid Samples		Plasma Samples	
	Positive	Negative	Positive	Negative
Homovanillic acid	28	0	28	0
Caffeic acid	27	1	28	0
3-hydroxyphenyl acetic acid	27	1	5	23
Dihydrocaffeic acid	0	28	17	11
Vanillic acid	0	28	10	18
Hippuric acid	0	28	17	11
3,4 dihydroxybenzoic acid	0	28	1	27

Other phenolics, such as 3-hydroxyhippuric, 4-hydroxyhippuric, chlorogenic acid, ellagic acid, and urolithin A, were not detected in CSF and plasma samples.

**Table 3 nutrients-12-01531-t003:** Concentrations of 3-hydroxyphenyl acetic acid (3HPAA), homovanillic acid (HVA) and caffeic acid (CA) in cerebrospinal fluid of patients subgroup with multiple sclerosis (*n* = 15) and subgroup with other neurological disorders (*n* = 13).

Phenolic Acid (µmol/L)	Multiple Sclerosis	Other Neurological Disorders
3HPAA	8.12 ± 10.08 (4.50; 5.39)	7.78 ± 7.18 (4.20; 8.32)
HVA	0.19 ± 0.13 (0.16; 0.15)	0.22 ± 0.13 (0.19; 0.17)
CA	0.02 ± 0.01 (0.03; 0.01)	0.02 ± 0.01 (0.02; 0.00)

No significant differences between subgroups were found (*p* > 0.05).

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
