# Peer review of "The Presence of Caffeic Acid in Cerebrospinal Fluid: Evidence That Dietary Polyphenols Can Cross the Blood-Brain Barrier in Humans"

_nutrients, 2020, doi:10.3390/nu12051531_

Round 1

Reviewer 1 Report

The present manuscript, submitted by Grabska-Kobylecka et al. determines the levels of 12 food-derived polyphenols in the cerebrospinal fluid and plasma of patients. While the premise of this small study is interesting there are considerable flaws in the presentation and conceptualization of this study that impact its value.

The time to last food-intake was not supplied, but it seems likely that patients did not eat between the taking of the blood sample (for which they were fasted) and the lumbar puncture 30 min later. The half-life of selected polyphenols in plasma should be considered and the impact on the findings should at the very least be discussed.

The figure in the manuscript is unreasonably large, as if trying to make up for the lack of further figures, which would be required. In fact, the manuscript needs a representative image of HPLC results and desparately needs a representation of the results discussed in section 3.2 in the form of a scatter blots. It seems likely that there is no correlation between plasma and CSF levels of CA because measured values are so small which seems to challenge the sensitivity of the assay

Comparisons between csf levels of HVA, CA and 3HPAA should not be performed and don’t provide any additional information. These data are unrelated to each other.

The N number should be identical in figures, if no compound was detected than the levels of that sample should be included as 0. Leaving samples that had no detectable compound out of the representation falsely magnifies the average levels of compound in plasma. – On a separate note it would be interesting to discuss the link between plasma 3HPAA levels and disease, is there a link between plasma levels and dysfunctional BBB?

Tables 1-3 are not included in the manuscript file nor in the supplementary

Author Response

Dear Reviewer,

Thank you very much for the reviews of our manuscript  “The presence of caffeic acid in cerebrospinal fluid: evidence that dietary polyphenols can cross the blood-brain barrier in humans”. Following the reviewers’ comments, we have made a number of changes in the revised version of the manuscript (marked in red). These changes have also been listed below.

Responses to the first reviewer’s comments:

1. The time to last food-intake was not supplied, but it seems likely that patients did not eat between the taking of the blood sample (for which they were fasted) and the lumber puncture 30 min later.

Response (Re):   

Blood and CSF samples were collected under fasting conditions between 8:30 and 9:30 a.m.

 Last meal (supper) was consumed about 15 hours before sampling. This is described in the section “Study protocol and patients”.

2. The half-life of selected polyphenols in plasma should be considered and the impact on the findings should at the very least be discussed.

Re:

This is discussed as follows at the end of the subsection “Limitation of the study”:

“Plasma and CSF samples were collected under fasting conditions after 15 hours from the last meal. Maximal increase in CA plasma levels was observed within 0.5-1 h after consumption of 200 ml of red wine containing 1.8 mg of CA [60]. Similarly, ingestion of 200 ml of instant coffee resulted in maximal increase in chlorogenic acid plasma levels  (an ester of caffeic acid and quinic acid) after 1 h  with the half-life time about 0.3 h [34]. These suggest that non-fasting plasma and CSF samples could have considerably increased CA concentrations. On the other hand, studies with ileostomy patients revealed that almost all ingested CA dose was absorbed in the small intestine, while in the case of chlorogenic acid it was only one third of the dose [61]. This indicates that majority of chlorogenic acid from food could reach the colon and be hydrolyzed by colon microbiota into free CA with its subsequent absorption into the blood. Thus, the absorption of CA can occur even after several hours from the last meal and the conditions of blood and CSF samples collection can have a rather moderate effect on CA measurement results. However, the considerable differences between CSF concentrations of CA and the remaining two phenolics (HVA and 3HPAA) are in line with these data and support, to same extent, the conclusion that the presence of caffeic acid in CSF results only from its passage through the blood-brain barrier.”

3. The figure in the manuscript is unreasonably large, as if trying to make up for the lack of further figures, which would be required. In fact, the manuscript needs a representative image of HPLC results and desperately needs a representation of the results discussed in section 3.2 in the form of a scatter plots.

Re:

Four additional figures have been added in the revised manuscript. Fig. 1 and Fig. 2 show representative chromatograms (HPLC-ECD and HPLC-UV-VIS)  of  CSF sample obtained from a single patient, while Fig. 4 and Fig. 5  illustrate the correlations (scatter plot) between plasma and CSF levels of caffeic acid and homovanillic acid.

4. It seems likely that there is no correlation between plasma and CSF levels of CA because measured values are so small which seems to challenge the sensitivity of the assay.

Re:

We agree with this hypothesis. This is inserted in the discussion of the revised manuscript.  

 5. Comparisons between CSF levels of HVA, CA and 3HPAA should not be performed and don’t provide any additional information. These data are unrelated to each other

Re:

HVA and 3HPAA are produced endogenously in the brain and in other tissues, and are also ingested with food. Caffeic acid derives only from food. CSF samples were collected under fasting conditions, 15 hours after the last meal. Higher concentrations of HVA and 3HPAA than the concentration of caffeic acid in CSF are in line with these data and support, to same extent, the conclusion that the presence of caffeic acid in CSF results from its passage through the blood-brain barrier. Therefore, we decided to leave these comparisons in the revised manuscript. However, this remark of the Reviewer  inspired us to additionally enhance the discussion (see the last sentence in the subsection “Limitation of the study”).  

 6. The N number should be identical in figures, if no compound was detected than the levels of that sample should be included as 0. Leaving samples that had no detectable compound out of the representation falsely magnifies the average levels of compound in plasma.

Re:

Figure 3 in the revised manuscript is the changed Figure 1 from the previous one. All calculations have been performed again following the reviewer’s suggestions, and the corresponding values of polyphenols concentrations in CSF and plasma (Fig. 3 , Table 3,  section Results and Abstract) have been changed.  

 7.On a separate note it would be interesting to discuss the link between plasma 3HPAA levels and disease, is there a link between plasma levels and dysfunctional BBB ?

Re :

This is discussed at the end of the subsection “3-hydroxyphenyl acetic acid  in CSF” as follows: “Interestingly, increased urinary excretion of 3HPAA was found in children with phenyloketonuria [48] and autistic patients [49] as a consequence of abnormal metabolism of phenylalanine. The overgrowth of intestinal microbiota such as Clostridium species may also lead to enhanced formation of 3HPAA [49]. Because some brain dysfunctions are observed in autistic patients and those with untreated phenyloketonuria, one may speculate that circulating 3HPAA may cross BBB and be toxic for neurons. On the other hand, in vitro experiments revealed a protective effect of 3HPAA on the BBB model against oxidative stress [50]. Therefore, as to whether 3HPAA is a biomarker of the afore-mentioned diseases or rather a metabolite involved in the development of brain damage is still an open question”.   

8. Tables 1-3 are not included in the manuscript file not in the supplementary

Re:

We uploaded these tables along with the manuscript, figure 1 and supplementary materials. Probably, due to some technical problems, the tables were not available for the Reviewers, for which we apologize. They are now available in the revised version of manuscript. Table 1 shows the basic characteristics of the studied patients and samples of cerebrospinal fluid. Table 2 shows the number of cerebrospinal fluid and plasma samples with detectable levels of studied polyphenols. Concentrations of  3-hydroxyphenyl acetic acid (3HPAA), homovanillic acid (HVA) and caffeic acid (CA) in cerebrospinal fluid of the subgroup of patients with multiple sclerosis and the subgroup with other neurological disorders is demonstrated in Table 3.

Responses to the second reviewer’s comments:

1. Although the Tables 1-3 are cited in the manuscript, there are no tables available for the review process. I cannot review Tables 1-3.

 Re:

We uploaded these tables along with the manuscript, figure 1 and supplementary materials. Probably, due to some technical problems, the tables were not available for the Reviewers, for which we apologize. They are now available in the revised version of manuscript. Table 1 shows the basic characteristics of the studied patients and samples of cerebrospinal fluid. Table 2 shows the number of cerebrospinal fluid and plasma samples with detectable levels of studied polyphenols. Concentrations of  3-hydroxyphenyl acetic acid (3HPAA), homovanillic acid (HVA) and caffeic acid (CA) in cerebrospinal fluid of the subgroup of patients with multiple sclerosis and the subgroup with other neurological disorders is demonstrated in Table 3.

 2. Figure 1: Some of the data shown have very large standard deviation. Wouldn’t it be better to use different units e.g nmol/L to get better graphs? The numbers in the graphs are not clear to me. Please explain the numbers in the figure caption.

 Re:

Figure 1 has been changed due to the calculations performed again following the first reviewer’s suggestions (response no. 6). The number (n=28) is the same for 3 polyphenols and is equal to the number of the studied patients. Standard deviation of some data is relatively large due to few individual results equal or close to 0.  Therefore, for data analysis we have used non-parametric tests.  This is now Fig. 3 in the revised version of manuscript.   

Due to these changes the number of figures increased by 4 and the one of cited references increased by 5.

Thank you very much for these comments. They have considerably improved quality of our manuscript.

Sincerely yours,

Dariusz Nowak

Reviewer 2 Report

The manuscript by Grabska-Kobylecka et al. shows valuable measurements of 12 polyphenols in plasma and cerebrospinal fluid of patients. The authors demonstrate for the first time, that the caffeic acid is present in the cerebrospinal fluid and therefore can cross the blood-brain barrier.

I have some comments:

Although the Tables 1-3 are cited in the manuscript, there are no tables available for the review process.

I cannot review Tables 1-3.

Figure 1: Some of the data shown have very large standard deviation. Wouldn’t it be better to use different units, e.g. nmol/L to get better graphs? The numbers in the graphs are not clear to me. Please explain the numbers in the figure caption.

Author Response

(The authors gave the same response as above.)

Round 2

Reviewer 1 Report

The research group around Grabska-Kobylecka et al. have done a very good job to update their manuscript "The presence of caffeic acid in cerebrospinal fluid: evidence that dietary polyphenols can cross the blood-brain barrier in humans" and taking into account the critical comments by the reviewers.

The overall manuscript and especially the presentation of the data has clearly improved. I would like to suggest some small changes/additions to the manuscript that could improve it further:

  • the discussion should contain a comment regarding the absence of most of the studied polyphenols from CSF and blood. Is the reason for the fact that e.g. cholorgenic acid and urolithin A were not detected in these samples only due to the long time since the last meal and the short half life of these polyphenols?
  • Figure 3 could be edited for purely aesthetic reasons, at the very least it would be nice to rearrange the fields containing average or mean data to not be on top of a gridline

Author Response

In response to the reviewer’s comments we have made a number of changes in the 2nd revised version of the manuscript  “The presence of caffeic acid in cerebrospinal fluid: evidence that dietary polyphenols can cross the blood-brain barrier in humans”. They have been all marked in red in the text of the article. They are listed below.

  1. The discussion should contain a comment regarding the absence of most of the studied polyphenols from CSF and blood. Is the reason for the fact that e.g. chlorogenic acid and urolithin A were not detected in these samples only due to the long time since the last meal and the short half-life of these polyphenols?

Response:

This is discussed at the end of the subsection “Limitation of the study” as follows:

“Other phenolic acids (e.g. ferulic acid, dihydrocaffeic acid, 3,4-dihydroxybenzoic acid) are also rapidly absorbed in the small intestine and reach maximal plasma concentration within 1 to 2 hours after ingestion. Their elimination lasts about a few hours [60]. Similarly, hippuric acids (hippuric acid, 3-hydroxyhippuric, 4-hydroxyhippuric), which are metabolites of various polyphenols, are rapidly excreted with the urine [60]. Therefore, it is possible that lack of detectable concentrations of these compounds in plasma and CSF samples could result from the relatively long fasting period preceding their collection.

This is in line with our previous observations [29]. In strawberry consumers (500 g of strawberry pulp daily, between 11:00 a.m. and 2:00 p.m. for 30 days) fasting plasma  had no detectable ellegic acid and its main metabolite urolithin A while HPLC analysis of spot morning urine revealed elevated concentrations of this last compound [29]. Nevertheless, an analysis of post-meal CSF and plasma samples would clearly solve this issue.“

  1. 2. Figure 3 could be edited for purely aesthetic reasons, at the very least it would be nice to rearrange the fields containing average or mean data to not be on top of a

Response:

Figure 3 has been improved according to the Reviewer’s suggestion.

Sincerely Yours,

Dariusz Nowak